# Permutation-Invariant Variational Autoencoder for Graph-Level Representation Learning

**Robin Winter**
Bayer AG
Freie Universität Berlin
`robin.winter@bayer.com`

**Frank Noé**
Freie Universität Berlin
`frank.noe@fu-berlin.de`

**Djork-Arné Clevert**
Bayer AG
`djork-arne.clevert@bayer.com`

## Abstract

Recently, there has been great success in applying deep neural networks on graph structured data. Most work, however, focuses on either node- or graph-level supervised learning, such as node, link or graph classification or node-level unsupervised learning (e.g., node clustering). Despite its wide range of possible applications, graph-level unsupervised representation learning has not received much attention yet. This might be mainly attributed to the high representation complexity of graphs, which can be represented by $n!$ equivalent adjacency matrices, where $n$ is the number of nodes. In this work we address this issue by proposing a permutation-invariant variational autoencoder for graph structured data. Our proposed model indirectly learns to match the node order of input and output graph, without imposing a particular node order or performing expensive graph matching. We demonstrate the effectiveness of our proposed model for graph reconstruction, generation and interpolation and evaluate the expressive power of extracted representations for downstream graph-level classification and regression.

## 1 Introduction

Graphs are an universal data structure that can be used to describe a vast variety of systems from social networks to quantum mechanics [1]. Driven by the success of Deep Learning in fields such as Computer Vision and Natural Language Processing, there has been an increasing interest in applying deep neural networks on non-Euclidean, graph structured data as well [2, 3]. Most notably, generalizing Convolutional Neural Networks and Recurrent Neural Networks to arbitrarily structured graphs for supervised learning has lead to significant advances on task such as molecular property prediction [4] or question-answering [5]. Research on unsupervised learning on graphs mainly focused on node-level representation learning, which aims at embedding the local graph structure into latent node representations [6, 7, 8, 9, 10]. Usually, this is achieved by adopting an autoencoder framework where the encoder utilizes e.g., graph convolutional layers to aggregate local information at a node level and the decoder is used to reconstruct the graph structure from the node embeddings. Graph-level representations are usually extracted by aggregating node-level features into a single vector, which is common practice in supervised learning on graph-level labels [4].

Unsupervised learning of graph-level representations, however, has not yet received much attention, despite its wide range of possible applications, such as feature extraction, pre-training for graph-level classification/regression tasks, graph matching or similarity ranking. This might be mainly attributed to the high representation complexity of graphs arising from their inherent invariance with respect

35th Conference on Neural Information Processing Systems (NeurIPS 2021).

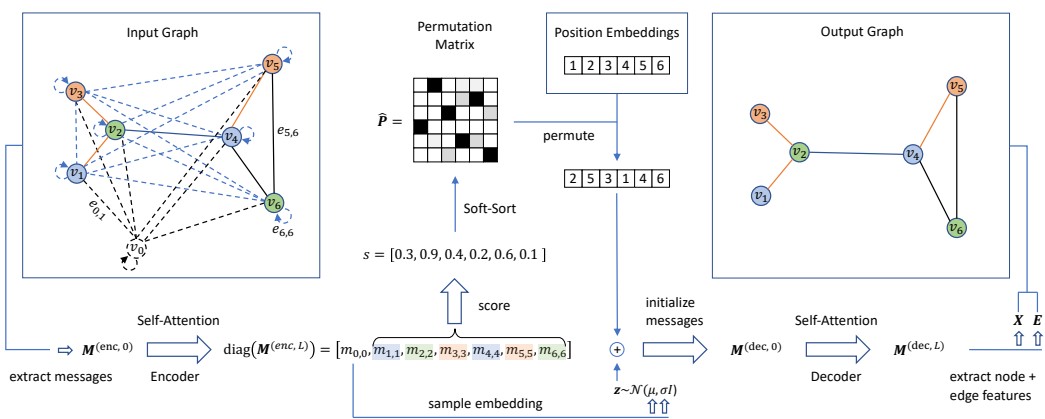

Figure 1: Network architecture of the proposed model. Input graph is depicted as fully connected graph (dashed lines for not direct neighbours) with the additional embedding node $v_0$ and edges to it (black color). Different node and edge types are represented by different colors and real edges by solid lines between nodes. Transformations parameterized by a neural network are represented by block arrows.

to the order of nodes within the graph. In general, a graph with $n$ nodes, can be represented by $n!$ equivalent adjacency matrices, each corresponding to a different node order. Since the general structure of a graph is invariant to the order of their individual nodes, a graph-level representation should not depend on the order of the nodes in the input representation of a graph, i.e. two isomorphic graphs should always be mapped to the same representation. This poses a problem for most neural network architectures which are by design not invariant to the order of their inputs. Even if carefully designed in a permutation invariant way (e.g., Graph Neural Networks with a final node aggregation step), there is no straight-forward way to train an autoencoder network, due to the ambiguous reconstruction objective, requiring the same discrete order of input and output graphs to compute the reconstruction loss.

How can we learn a permutation-invariant graph-level representation utilizing a permutation-variant reconstruction objective? In this work we tackle this question proposing a graph autoencoder architecture that is by design invariant to the order of nodes in a graph. We address the order ambiguity issue by training alongside the encoder and decoder model an additional *permuter* model that assigns to each input graph a permutation matrix to align the input graph node order with the node order of the reconstructed graph.

## 2 Method

### 2.1 Notations and Problem Definition

An undirected Graph $\mathcal{G} = (\mathcal{V}, \mathcal{E})$ is defined by the set of $n$ nodes $\mathcal{V} = \{v_1, \ldots, v_n\}$ and edges $\mathcal{E} = \{(v_i, v_j)|v_i, v_j \in \mathcal{V}\}$. We can represent a graph in matrix form by its node features $\mathbf{X}_\pi \in \mathbb{R}^{n \times d_v}$ and adjacency matrix $\mathbf{A}_\pi \in \{0, 1\}^{n \times n}$ in the node order $\pi \in \Pi$, where $\Pi$ is the set of all $n!$ permutations over $\mathcal{V}$. We define the permutation matrix $\mathbf{P}$ that reorders nodes from order $\pi$ to order $\pi'$ as $\mathbf{P}_{\pi \to \pi'} = (p_{ij}) \in \{0, 1\}^{n \times n}$, with $p_{ij} = 1$ if $\pi(i) = \pi'(j)$ and $p_{ij} = 0$ everywhere else. Since Graphs are invariant to the order of their nodes, note that

$$\mathcal{G}_\pi = \mathcal{G}(\mathbf{X}_\pi, \mathbf{A}_\pi) = \mathcal{G}(\mathbf{P}_{\pi \to \pi'}\mathbf{X}_\pi, \mathbf{P}_{\pi \to \pi'}\mathbf{A}_\pi \mathbf{P}_{\pi \to \pi'}^\top) = \mathcal{G}(\mathbf{X}_{\pi'}, \mathbf{A}_{\pi'}) = \mathcal{G}_{\pi'}, \qquad (1)$$

where $\top$ is the transpose operator. Let us now consider a dataset of graphs $\mathbf{G} = \{\mathcal{G}^{(i)}\}_{i=0}^N$ we would like to represented in a low-dimensional continuous space. We can adopt a latent variable approach and assume that the data is generate by a process $p_\theta(\mathcal{G}|\mathbf{z})$, involving an unobserved continuous random variable $\mathbf{z}$. Following the work of Kingma and Welling [11], we approximate the intractable posterior by $q_\phi(\mathcal{G}|\mathbf{z}) \approx p_\theta(\mathcal{G}|\mathbf{z})$ and minimize the lower bound on the marginal likelihood of graph $\mathcal{G}^{(i)}$:

$$\log p_\theta(\mathcal{G}^{(i)}) \geq \mathcal{L}(\phi, \theta; \mathcal{G}^{(i)}) = -\mathrm{KL}\left[q_\phi(\mathbf{z}|\mathcal{G}^{(i)})||p_\theta(\mathbf{z})\right] + \mathbb{E}_{q_\phi(\mathbf{z}|\mathcal{G}^{(i)})}\left[\log p_\theta(\mathcal{G}^{(i)}|\mathbf{z})\right], \qquad (2)$$

where the Kullback–Leibler (KL) divergence term regularizes the encoded latent codes of graphs $\mathcal{G}^{(i)}$ and the second term enforces high similarity of decoded graphs to their encoded counterparts. As graphs can be completely described in matrix form by their node features and adjacency matrix, we can parameterize $q_\phi$ and $p_\theta$ in Eq. (2) by neural networks that encode and decode node features $\mathbf{X}_\pi^{(i)}$ and adjacency matrices $\mathbf{A}_\pi^{(i)}$ of graphs $\mathcal{G}_\pi^{(i)}$. However, as graphs are invariant under arbitrary node re-ordering, the latent code $\mathbf{z}$ should be invariant to the node order $\pi$:

$$q_\phi(\mathbf{z}|\mathcal{G}_\pi) = q_\phi(\mathbf{z}|\mathcal{G}_{\pi'}), \text{ for all } \pi, \pi' \in \Pi. \tag{3}$$

This can be achieved by parameterizing the encoder model $q_\phi$ by a permutation invariant function. However, if the latent code $\mathbf{z}$ does not encode the input node order $\pi$, input graph $\mathcal{G}_\pi$ and decoded graph $\hat{\mathcal{G}}_{\pi'}$ are no longer necessarily in the same order, as the decoder model has no information about the node order of the input graph. Hence, the second term in Eq. (2) cannot be optimized anymore by minimizing the reconstruction loss between encoded graph $\mathcal{G}_\pi$ and decoded graph $\hat{\mathcal{G}}_{\pi'}$ in a straight-forward way. They need to be brought in the same node order first. We can rewrite the expectation in Eq. (2) using Eq. (1):

$$\mathbb{E}_{q_\phi(\mathbf{z}|\mathcal{G}^{(i)})} \left[ \log p_\theta(\mathcal{G}_{\pi'}^{(i)}|\mathbf{z}) \right] = \mathbb{E}_{q_\phi(\mathbf{z}|\mathcal{G}^{(i)})} \left[ \log p_\theta(\hat{\mathbf{P}}_{\pi \to \pi'} \mathcal{G}_\pi^{(i)}|\mathbf{z}) \right]. \tag{4}$$

Since the ordering of the decoded graph $\pi'$ is subject to the learning process of the decoder and thus unknown in advance, finding $\mathbf{P}_{\pi \to \pi'}$ is not trivial. In [12], the authors propose to use approximate graph matching to find the permutation matrix $\mathbf{P}_{\pi \to \pi'}$ that maximizes the similarity $s(X_{\pi'}, \mathbf{P}_{\pi \to \pi'} \hat{X}_\pi; A_{\pi'}, \mathbf{P}_{\pi \to \pi'} \hat{A}_\pi \mathbf{P}_{\pi \to \pi'}^\top)$, which involves up to $O(n^4)$ re-ordering operations at each training step in the worst case [13].

## 2.2 Permutation-Invariant Variational Graph Autoencoder

In this work we propose to solve the reordering problem in Eq. (4) implicitly by inferring the permutation matrix $\mathbf{P}_{\pi' \to \pi}$ from the input graph $\mathcal{G}_\pi$ by a model $g_\psi(\mathcal{G}_\pi)$ that is trained to bring input and output graph in the same node order and is used by the decoder model to permute the output graph. We train this *permuter* model jointly with the *encoder* model $q_\phi(\mathbf{z}|\mathcal{G}_\pi)$ and *decoder* model $p_\theta(\mathcal{G}_\pi|\mathbf{z}, \mathbf{P}_{\pi' \to \pi})$, optimizing:

$$\mathcal{L}(\phi, \theta, \psi; \mathcal{G}_\pi^{(i)}) = -\text{KL}\left[ q_\phi(\mathbf{z}|\mathcal{G}_\pi^{(i)})||p_\theta(\mathbf{z}) \right] + \mathbb{E}_{q_\phi(\mathbf{z}|\mathcal{G}^{(i)})} \left[ \log p_\theta(\mathcal{G}_\pi^{(i)}|\mathbf{z}, g_\psi(\mathcal{G}_\pi^{(i)})) \right]. \tag{5}$$

Intuitively, the permuter model has to learn how the ordering of nodes in the graph generated by the decoder model will differ from a specific node order present in the input graph. During the learning process, the decoder will learn its own canonical ordering that, given a latent code $z$, it will always reconstruct a graph in. The permuter learns to transform/permute this canonical order to a given input node order. For this, the permuter predicts for each node $i$ of the input graph a score $s_i$ corresponding to its probability to have a low node index in the decoded graph. By sorting the input nodes indices by their assigned scores we can infer the output node order and construct the respective permutation matrix $\mathbf{P}_{\pi \to \pi'} = (p_{ij}) \in \{0, 1\}^{n \times n}$, with

$$p_{ij} = \begin{cases} 1, & \text{if } j = \text{argsort}(s)_i \\ 0, & \text{else} \end{cases} \tag{6}$$

to align input and output node order. Since the argsort operation is not differentiable, we utilizes the continuous relaxation of the argsort operator proposed in [14, 15]:

$$\mathbf{P} \approx \hat{\mathbf{P}} = \text{softmax}(\frac{-d(\text{sort}(s)\mathbb{1}^\top, \mathbb{1}s^\top)}{\tau}), \tag{7}$$

where the softmax operator is applied row-wise, $d(x, y)$ is the $L_1$-norm and $\tau \in \mathbb{R}_+$ a temperature-parameter. By utilizing this continuous relaxation of the argsort operator, we can train the permuter model $g_\psi$ in Eq. (5) alongside the encoder and decoder model with stochastic gradient descent. In order to push the relaxed permutation matrix towards a real permutation matrix (only one 1 in every row and column), we add to Eq. (5) a row- and column-wise entropy term as additional penalty term:

$$C(\mathbf{P}) = \sum_i H(\bar{\mathbf{p}}_{i,\cdot}) + \sum_j H(\bar{\mathbf{p}}_{\cdot,j}), \tag{8}$$

with Shannon entropy $H(x) = -\sum_i x_i \log(x_i)$ and normalized probabilities $\overline{\mathbf{p}}_{i,\cdot} = \frac{\mathbf{p}_{i,\cdot}}{\sum_j \mathbf{p}_{i,j}}$.

**Propositions 1.** *A square matrix $\mathbf{P}$ is a real permutation matrix if and only if $C(\mathbf{P}) = 0$ and the doubly stochastic constraint $p_{ij} \geq 0 \ \forall(i,j), \ \sum_i p_{ij} = 1 \ \forall j, \ \sum_j p_{ij} = 1 \ \forall i$ holds.*

**Proof.** *See Appendix A.*

By enforcing $\hat{\mathbf{P}} \to \mathbf{P}$, we ensure that no information about the graph structure is encoded in $\hat{\mathbf{P}}$ and decoder model $p_\theta(\mathcal{G}_\pi|\mathbf{z}, \mathbf{P})$ can generate valid graphs during inference, without providing a specific permutation matrix $\mathbf{P}$ (e.g., one can set $\mathbf{P} = \mathbf{I}$ and decode the learned canonical node order). At this point it should also be noted, that our proposed framework can easily be generalized to arbitrary sets of elements, although we focus this work primarily on sets of nodes and edges defining a graph.

**Graph Isomorphism Problem.** Equation (5) gives us means to train an autoencoder framework with a permutation invariant encoder that maps a graph $f : \mathcal{G} \to \mathcal{Z}$ in an efficient manner. Such an encoder will always map two topologically identical graphs (even with different node order) to the same representation $z$. Consequently, the question arises, if we can decide for a pair of graphs whether they are topologically identical. This is the well-studied *graph isomorphism problem* for which no polynomial-time algorithm is known yet [16, 17]. As mentioned above, in our framework, two isomorphic graphs will always be encoded to the same representation. Still, it might be that two non-isomorphic graphs will be mapped to the same point (non-injective). However, if the decoder is able to perfectly reconstruct both graphs (which is easy to check since the permuter can be used to bring the decoded graph in the input node order), two non-isomorphic graphs must have a different representation $z$. If two graphs have the same representation and the reconstruction fails, the graphs might still be isomorphic but with no guarantees. Hence, our proposed model can solve the graph isomorphism problem at least for all graphs it can reconstruct.

## 2.3 Details of the Model Architecture

In this work we parameterize the encoder, decoder and permuter model in Eq. (5) by neural networks utilizing the self-attention framework proposed by Vaswani et al. [18] on directed messages representing a graph. Figure 1, visualizes the architecture of the proposed permutation-invariant variational autoencoder. In the following, we describe the different parts of the model in detail[1].

**Graph Representation by Directional Messages.** In general, most graph neural networks can be thought of as so called Message Passing Neural Networks (MPNN) [19]. The key idea of MPNNs is the aggregation of neighbourhood information by passing and receiving messages of each node to and from neighbouring nodes in a graph. We adopt this view and represent graphs by its messages between nodes. We represent a graph $\mathcal{G}(\mathbf{X}, \mathbf{E})$, with node features $\mathbf{X} \in \mathbb{R}^{n \times d_v}$ and edge features $\mathbf{E} \in \mathbb{R}^{n \times n \times d_e}$, by its message matrix $\mathbf{M} = (\mathbf{m}_{ij}) \in \mathbb{R}^{n \times n \times d_m}$:

$$\mathbf{m}_{ij} = \sigma\left([\mathbf{x}_i||\mathbf{x}_j||\mathbf{e}_{ij}]\mathbf{W} + \mathbf{b}\right), \tag{9}$$

with non-linearity $\sigma$, concatenation operator $||$ and trainable parameters $\mathbf{W}$ and $\mathbf{b}$. Note, that nodes in this view are represented by *self-messages* $\mathrm{diag}(\mathbf{M})$, messages between non-connected nodes exists, although the presence or absence of a connection might be encoded in $\mathbf{e}_{ij}$, and if $\mathbf{M}$ is not symmetric, edges have an inherent direction.

**Self-Attention on Directed Messages.** We follow the idea of aggregating messages from neighbours in MPNNs, but utilize the self-attention framework proposed by Vaswani et al. [18] for sequential data. Our proposed model comprises multiple layers of multi-headed scaled-dot product attention. One attention head is defined by:

$$\mathrm{Attention}\left(\mathbf{Q}, \mathbf{K}, \mathbf{V}\right) = \mathrm{softmax}\left(\frac{\mathbf{Q}\mathbf{K}^\top}{\sqrt{d_k}}\right)\mathbf{V} \tag{10}$$

with queries $\mathbf{Q} = \mathbf{M}\mathbf{W}^Q$, keys $\mathbf{K} = \mathbf{M}\mathbf{W}^K$, and values $\mathbf{V} = \mathbf{M}\mathbf{W}^V$ and trainable weights $\mathbf{W}^Q \in \mathbb{R}^{d_m \times d_q}$, $\mathbf{W}^K \in \mathbb{R}^{d_m \times d_k}$ and $\mathbf{W}^V \in \mathbb{R}^{d_m \times d_v}$. For multi-headed self-attention we concatenate multiple attention heads together and feed them to a linear layer with $d_m$ output features. Since the message matrix $\mathbf{M}$ of a graph with $n$ nodes comprises $n^2$ messages, attention of all messages to all messages would lead to a $O(n^4)$ complexity. We address this problem by letting messages

---

[1] Code available at `https://github.com/jrwnter/pigvae`

$\mathbf{m}_{ij}$ only attend on incoming messages $\mathbf{m}_{ki}$, reducing the complexity to $O(n^3)$. We achieve this by representing $\mathbf{Q}$ as a $(m \times n \times d)$ tensor and $\mathbf{K}$ and $\mathbf{V}$ by a transposed $(n \times m \times d)$ tensor, resulting into a $(m \times n \times m)$ attention tensor, with number of nodes $m = n$ and number of features $d$. That way, we can efficiently utilize batched matrix multiplications in Eq. (10), in contrast to computing the whole $(n^2 \times n^2)$ attention matrix and masking attention on not incoming messages out.

**Encoder**  To encode a graph into a fixed-sized, permutation-invariant, continuous latent representation, we add to input graphs a dummy node $v_0$, acting as an embedding node. To distinguish the embedding node from other nodes, we add an additional node and edge type to represent this node and edges to and from this node. After encoding this graph into a message matrix $\mathbf{M}^{(\text{enc, }0)}$ as defined in Eq. (9), we apply $L$ iterations of self-attention to update $\mathbf{M}^{(\text{enc, }L)}$, accumulating the graph structure in the embedding node, represented by the self-message $\mathbf{m}_{0,0}^{(\text{enc, }L)}$. Following [11], we utilize the reparameterization trick and sample the latent representation $\mathbf{z}$ of a graph by sampling from a multivariate normal distribution:

$$\mathbf{z} \sim \mathcal{N}(f_\mu(\mathbf{m}_{0,0}^{(\text{enc, }L)}), f_\sigma(\mathbf{m}_{0,0}^{(\text{enc, }L)})\mathbf{I}), \tag{11}$$

with $f_\mu : \mathbf{m}_{0,0} \to \mu \in \mathbb{R}^{d_z}$ and $f_\sigma : \mathbf{m}_{0,0} \to \sigma \in \mathbb{R}^{d_z}$, parameterized by a linear layer.

**Permuter**  To predict how to re-order the nodes in the output graph to match the order of nodes in the input graph, we first extract node embeddings represented by self-messages on the main diagonal of the encoded message matrix $\mathbf{m}_{i,i}^{(\text{enc, }L)} = \text{diag}(\mathbf{M}^{(\text{enc, }L)})$ for $i > 0$. We score these messages by a function $f_s : \mathbf{m}_{i,i} \to s \in \mathbb{R}$, parameterized by a linear layer and apply the soft-sort operator (see Eq. (7)) to retrieve the permutation matrix $\hat{\mathbf{P}}$.

**Decoder**  We initialize the message matrix for the decoder models input with the latent representation $\mathbf{z}$ at each entry. To break symmetry and inject information about the relative position/order of nodes to each other, we follow [18] and define position embeddings in dimension $k$

$$\text{PE}(i)_k = \begin{cases} \sin(i/10000^{2k/d_z}), & \text{for even } k \\ \cos(i/10000^{2k/d_z}), & \text{for odd } k \end{cases} \tag{12}$$

It follows for the initial decoder message matrix $\mathbf{M}^{(\text{dec, }0)}$:

$$\mathbf{m}_{ij}^{(\text{dec, }0)} = \sigma\left([\mathbf{z} + [\text{PE}(i)||\text{PE}(j)]]\,\mathbf{W} + \mathbf{b}\right), \tag{13}$$

Since the self-attention based decoder model is permutation equivariant, we can move the permutation operation in Eq. (5) in front of the decoder model and directly apply it to the position embedding sequence (see Figure 1). After $L$ iterations of self-attention on the message matrix $\mathbf{M}$, we extract node features $\mathbf{x}_i \in \mathbf{X}$ and edge features $\mathbf{e}_{i,j} \in \mathbf{E}$ by a final linear layer:

$$\mathbf{x}_i = \mathbf{m}_{i,i}\mathbf{W}^v + \mathbf{b}^v \qquad \mathbf{e}_{i,j} = 0.5 \cdot (\mathbf{m}_{i,j} + \mathbf{m}_{j,i})\mathbf{W}^e + \mathbf{b}^e, \tag{14}$$

with learnable parameters $\mathbf{W}^v \in \mathbb{R}^{d_m \times d_v}$, $\mathbf{W}^e \in \mathbb{R}^{d_m \times d_e}$, $\mathbf{b}^v \in \mathbb{R}^{d_v}$ and $\mathbf{b}^e \in \mathbb{R}^{d_e}$.

**Overall Architecture**  We now describe the full structure of our proposed method using the ingredients above (see Figure 1). Initially, the input graph is represented by the directed message matrix $\mathbf{M}^{(\text{enc, }0)}$, including an additional graph embedding node $v_0$. The encoder model performs $L$ iterations of self-attention on incoming messages. Next, diagonal entries of the resulting message matrix $\mathbf{M}^{(\text{enc, }L)}$ are extracted. Message $m_{0,0}^{(\text{enc, }L)}$, representing embedding node $v_0$, is used to condition the normal distribution, graph representation $\mathbf{z}$ is sampled from. The other diagonal entries $m_{i,i}^{(\text{enc, }L)}$ are transformed into scores and sorted by the Soft-Sort operator to retrieve the permutation matrix $\hat{\mathbf{P}}$. Next, position embeddings (in Figure 1 represented by single digits) are re-ordered by applying $\hat{\mathbf{P}}$ and added by the sampled graph embedding $\mathbf{z}$. The resulting node embeddings are used to initialize message matrix $\mathbf{M}^{(\text{dec, }0)}$ and fed into the decoding model. After $L$ iterations of self-attention, diagonal entries are transformed to node features $\mathbf{X}$ and off-diagonal entries to edge features $\mathbf{E}$ to generate the output graph. In order to train and infer on graphs of different size, we pad all graphs in a batch with empty nodes to match the number of nodes of the largest graph. Attention on empty nodes is masked out at all time. To generate graphs of variable size, we train alongside the variational autoencoder an additional multi-layer perceptron to predict the number of atoms of graph from its latent representation $\mathbf{z}$. During inference, this model informs the decoder on how many nodes to attend to.

Table 1: Negative log likelihood (NLL) and area under the receiver operating characteristics curve (ROC-AUC) for reconstruction of the adjacency matrix of graphs from different families. We compare our proposed method (PIGAE) with Graph Autoencoder (GAE) [8] and results of Graphite and Graph Autoencoder (GAE*) reported in [20]. PIGAE* utilize topological distances of nodes in a graph as edge feature.

| Models | Erdos-Renyi | | Barabasi-Albert | | Ego | |
|---|---|---|---|---|---|---|
| | NLL | ROC-AUC | NLL | ROC-AUC | NLL | ROC-AUC |
| PIGAE | $20.5 \pm 0.9$ | $98.3 \pm 0.1$ | $27.2 \pm 0.9$ | $96.7 \pm 0.2$ | $23.4 \pm 0.5$ | $97.8 \pm 0.3$ |
| **PIGAE*** | $\mathbf{19.5 \pm 0.8}$ | $\mathbf{99.4 \pm 0.1}$ | $\mathbf{15.2 \pm 0.8}$ | $\mathbf{99.5 \pm 0.1}$ | $\mathbf{22.4 \pm 0.5}$ | $\mathbf{98.8 \pm 0.3}$ |
| GAE | $186 \pm 3$ | $57.9 \pm 0.1$ | $199 \pm 3$ | $57.4 \pm 0.1$ | $191 \pm 4$ | $59.1 \pm 0.1$ |
| GAE* | $222 \pm 8$ | - | $236 \pm 15$ | - | $197 \pm 2$ | - |
| Graphite | $196 \pm 1$ | - | $192 \pm 2$ | - | $183 \pm 1$ | - |

**Key Architectural Properties** Since no position embeddings are added to the input of the encoders self-attention layers, accumulated information in the single embedding node $v_0$ ($\mathbf{m}_{0,0}$) is invariant to permutations of the input node order. Hence, the resulting graph embedding $\mathbf{z}$ is permutation invariant as well. This is in stark contrast to classical graph autoencoder frameworks [8, 20, 21], that encode whole graphs effectively by concatenating all node embeddings, resulting in a graph-level representation that is different for isomorphic graphs, as the sequence of node embeddings permutes equivalently with the input node order. As no information about the node order is encoded in the graph embedding $\mathbf{z}$, the decoder learns its own (canonical) node order, distinct graphs are deterministically decoded in. The input node order does not influence this decoded node order. As the decoder is based on permutation equivariant self-attention layers, this canonical order is solely defined with respect to the sequence of position embeddings used to initialize the decoders input. If the sequence of position embeddings is permuted, the decoded node order permutes equivalently. Thus, by predicting the right permutation matrix, input and output order can be aligned to correctly calculate the reconstruction loss. Input to the permuter model $[\mathbf{m}_{1,1}, \ldots, \mathbf{m}_{n+1,n+1}]$ is equivariant to permutations in the input node order (due to the equivariant self-attention layers in the encoder). Since the permuter model itself (i.e., the scoring function) is also permutation equivariant (node-wise linear layer), resulting permutation matrices $\mathbf{P}$ are equivariant to permutations in the input node order. Consequently, if the model can correctly reconstruct a graph in a certain node order, it can do it for all $n!$ input node orders, and the learning process of the whole model is independent to the node order of graphs in the training set.

## 3 Related Works

Most existing research on unsupervised graph representation learning focuses on *node-level* representation learning and can be broadly categorized in either shallow methods based on matrix factorization [22, 23, 24, 25] or random walks [26, 27], and deep methods based on Graph Neural Networks (GNN) [2, 3]. Kipf and Welling [8] proposed a graph autoencoder (GAE), reconstructing the adjacency matrix by taking the dot product between the latent node embeddings encoded by a GNN. Grover et al. [20] build on top of the GAE framework by parameterizing the decoder with additional GNNs, further refining the decoding process. Although *graph-level* representations in GAE-like approaches can be constructed by concatenating all node-level representations, note, that as a consequence they are only permutation equivariant and not permutation invariant. Permutation invariant representations could be extracted only after training by aggregating node embeddings into a single-vector representation. However, such a representation might miss important global graph structure. Samanta et al. [21] proposed a GAE-like approach, which parameters are trained in a permutation invariant way, following [28] utilizing breadth-first-traversals with randomized tie breaking during the child-selection step. However, as graph-level representations are still constructed by concatenation of node-level embeddings, this method still encodes graphs only in a permutation-equivariant way. A different line of work utilized Normalizing Flows [29] to address variational inference on graph structured data based on node-level latent variables [30, 31, 32].

Research on *graph-level* representations has mainly focused on supervised learning, e.g., graph-level classification by applying a GNN followed by a global node feature aggregation step (e.g., mean or max pooling) [4] or a jointly learned aggregation scheme [33]. Research on graph-level unsupervised

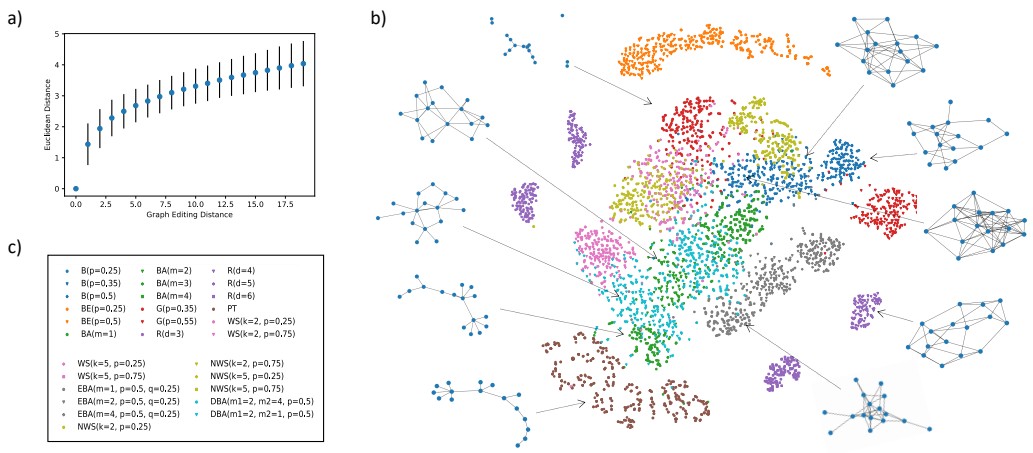

Figure 2: a) Euclidean distance over graph editing distance, averaged over 1000 Barabasi-Albert graphs with $m = 2$. b) t-SNE projection of representations from ten different graph families with different parameters. Example graphs are shown for some of the clusters. c) Legend of t-SNE plot explaining colours and symbols. Graph family abbreviations: Binomial (B), Binomial Ego (BE), Barabasi-Albert (BA), Geometric (G), Regular (R), Powerlaw Tree (PT), Watts-Strogatz (WA), Extended Barabasi-Albert (EBA), Newman-Watts-Strogatz (NWA), Dual-Barabasi-Albert (DBA).

representation learning has not yet received much attention and existing work is mainly based on contrastive learning approaches. Narayanan et al. [34] adapted the *doc2vec* method from the field of natural language processing to represent whole graphs by a fixed size embedding, training a *skipgram* method on rooted subgraphs (*graph2vec*). Bai et al. [35] proposed a *Siamese* network architecture, trained on minimizing the difference between the Euclidean distance of two encoded graphs and their graph editing distance. Recently, Sun et al. [36], adapted the *Deep InfoMax* architecture [37] to graphs, training on maximizing the mutual information between graph-level representations and representations of sub-graphs of different granularity (*InfoGraph*). Although those contrastive learning approaches can be designed to encode graphs in a permutation invariant way, they cannot be used to reconstruct or generate graphs from such representations.

Another line of related work concerns itself with *generative models* for graphs. Besides methods based on variational autoencoders [8, 20, 12] and Normalizing Flows [30, 31, 32], graph generative models have also been recently proposed based on generative adversarial neural networks [38, 39] and deep auto-regressive models [40, 28, 41]. Moreover, due to its high practical value for drug discovery, many graph generating methods have been proposed for molecular graphs [42, 43, 44, 45, 21]. Although graph generative models can be trained in a permutation invariant way [42, 21], those models can not be used to extract permutation invariant graph-level representations.

Recently, Yang et al. [46] proposed a GAE-like architecture with a node-feature aggregation step to extract permutation invariant graph-level representations that can also be used for graph generation. They tackle the discussed ordering issue of GAEs in the reconstruction by training alongside the GAE a Generative Adversarial Neural Network, which's permutation invariant discriminator network is used to embed input and output graph into a latent space. That way, a permutation invariant reconstruction loss can be defined as a distance in this space. However, as this procedure involves adversarial training of the reconstruction metric, this only approximates the exact reconstruction loss used in our work and might lead to undesirable graph-level representations.

## 4   Experimental Evaluation

We perform experiments on synthetically generated graphs and molecular graphs from the public datasets QM9 and PubChem. At evaluation time, predicted permutation matrices are always discretized to ensure their validity. For more details on training, see Appendix C.

Table 2: Classification Accuracy of our method (PIGAE), classical GAE, InfoGraph (IG), Shortest Path Kernel (SP) and Weisfeiler-Lehman Sub-tree Kernel (WL) on graph class prediction.

| PIGAE | GAE | IG | SP | WL |
|---|---|---|---|---|
| $\mathbf{0.83 \pm 0.01}$ | $0.65 \pm 0.01$ | $0.75 \pm 0.02$ | $0.50 \pm 0.02$ | $0.73 \pm 0.01$ |

## 4.1 Synthetic Data

**Graph Reconstruction**   In the first experiment we evaluate our proposed method on the reconstruction performance of graphs from graph families with a well-defined generation process. Namely, Erdos-Renyi graphs [47], with an edge probability of $p = 0.5$, Barabasi-Albert graphs [48], with $m = 4$ edges preferentially attached to nodes with high degree and Ego graphs. For each family we uniformly sample graphs with 12-20 nodes. The graph generation parameters match the ones reported in [20], enabling us to directly compare to Graphite. As additional baseline we compare against the Graph Autoencoder (GAE) proposed by Kipf and Welling [8]. As Grover et al. [20] only report negative log-likelihood estimates for their method Graphite and baseline GAE we also reevaluate GAE and report both negative log-likelihood (NLL) estimates for GAE to make a better comparison to Graphite possible (accounting for differences in implantation or the graph generation process). In Table 1 we show the evaluation metrics on a fixed test set of 25000 graphs for each graph family. On all four graph datasets our proposed model significantly outperforms the baseline methods, reducing the NLL error in three of the four datasets by approximately one magnitude. Utilizing the topological distance instead of just the connectivity as edge feature (compare [49]) further improves the reconstruction performance.

**Qualitative Evaluation**   To evaluate the representations learned by our proposed model, we trained a model on a dataset of randomly generated graphs with variable number of nodes from ten different graph families with different ranges of parameters (see Appendix B for details). Next, we generated a test set of graphs with a fixed number of nodes from these ten different families and with different fixed parameters. In total we generated graphs in 29 distinct settings. In Figure 2, we visualized the t-SNE projection [50] of the graph embeddings, representing different families by colour and different parameters within each family by different symbols. In this 2-D projection, we can make out distinct clusters for the different graph sets. Moreover, clusters of similar graph sets tend to cluster closer together. For example, Erdos-Renyi graphs form for each edge probability setting (0.25, 0.35, 0.5) a distinct cluster, while clustering in close proximity. As some graph families with certain parameters result in similar graphs, some clusters are less separated or tend to overlap. For example, the Dual-Barabasi-Albert graph family, which attaches nodes with either $m_1$ or $m_2$ other nodes, naturally clusters in between the two Barabasi-Albert graph clusters with $m = m_1$ and $m = m_2$.

**Graph Editing Distance**   A classical way of measuring graph similarity is the so called *graph editing distance* (GED) [51]. The GED between two graphs measures the minimum number of graph editing operations to transform one graph into the other. The set of operations typically includes inclusion, deletion and substitution of nodes or edges. To evaluate the correlation between similarity in graph representation and graph editing distance, we generated a set of 1000 Barabasi-Albert ($m = 3$) graphs with 20 nodes. For each graph we successively substitute randomly an existing edge by a new one, creating a set of graphs with increasing GED with respect to the original graph. In Figure 2, we plot the mean Euclidean distance between the root graphs and their 20 derived graphs with increasing GED. We see a strong correlation between GED and Euclidean distance of the learned representations. In contrast to classical GAEs, random permutations of the edited graphs have no effect on this correlation (see Appendix D for comparison).

**Graph Isomorphism and Permutation Matrix**   To empirical analyse if our proposed method detects isomorphic graphs, we generated for 10000 Barabasi-Albert graphs with up to 28 nodes a randomly permuted version and a variation only one graph editing step apart. For all graphs the Euclidean distance between original graph and edited graph was at least greater than $0.3$. The randomly permuted version always had the same embedding. Even for graphs out of training domain (geometric graphs with 128 nodes) all isomorphic and non-isomorphic graphs could be detected. Additionally, we investigated how well the permuter model can assign a permutation matrix to graph.

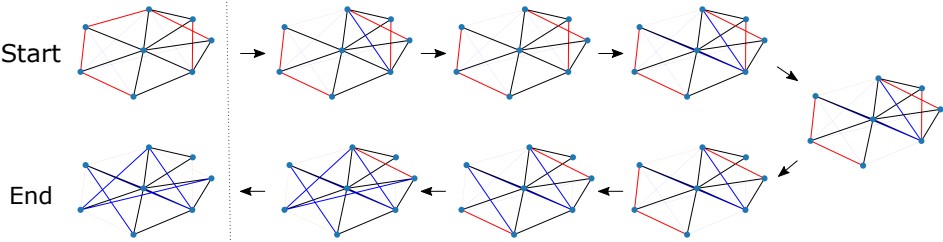

Figure 3: Linear interpolation between two graphs in the embedding space. Start graph's edges are colored red. End graph's edges are colored blue. Edges present in both graphs are colored black. Thick lines are present in a decoded graph, thin lines are absent.

As the high reconstruction performance in Table 1 suggest, most of the time, correct permutation matrices are assigned (See Appendix E for a more detailed analysis). We find that permutation matrices equivalently permute with the permutation of the input graph (See Appendix E).

**Graph Classification**   In order to quantitatively evaluate how meaningful the learned representations are, we evaluate the classification performance of a Support Vector Machine (SVM) on predicting the correct graph set label (29 classes as defined above) from the graph embedding. As baseline we compare against SVMs based on two classical graph kernel methods, namely Shortest Path Kernel (SP) [52] and Weisfeiler-Lehman Sub-tree Kernel (WL) [53] as well as embeddings extracted by the recently proposed contrastive learning model InfoGraph (IG) [36] and averaged node embeddings extracted by a classical GAE. Our model, IG and GAE where trained on the same synthetic dataset. In Table 2 we report the accuracy for each model and find the SVM based on representations from our proposed model to significantly outperform all baseline models. Notably, representations extracted from a classical GAE model, by aggregating (averaging) node-level embeddings into a graph-level embedding, perform significantly worse compared to representations extracted by our method. This finding is consistent with our hypothesis that aggregation of unsupervised learned node-level (local) features might miss important global features, motivating our work on graph-level unsupervised representation learning.

**Graph Interpolation**   The permutation invariance property of graph-level representations also enables the interpolation between two graphs in a straight-forward way. With classical GAEs such interpolations cannot be done in a meaningful way, as interpolations between permutation dependent graph embeddings would affect both graph structure as well as node order. In Figure 3 we show how successive linear interpolation between the two graphs in the embedding space results in smooth transition in the decoded graphs, successively deleting edges from the start graph (red) and adding edges from the end graph (blue). To the best of our knowledge, such graph interpolations have not been report in previous work yet and might show similar impact in the graph generation community as interpolation of latent spaces did in the field of Natural Language Processing and Computer Vision.

## 4.2   Molecular Graphs

Next, we evaluate our proposed model on molecular graphs from the QM9 dataset [54, 55]. This datasets contains about 134 thousand organic molecules with up to 9 heavy atoms (up to 29 atoms/nodes including Hydrogen). Graphs have 5 different atom types (C, N, O, F and H), 3 different formal charge types (-1, 0 and 1) and 5 differ bond types (no-, single-, double-, triple- and aromatic bond). Moreover, the dataset contains an energetically favorable conformation for each molecule in form of Cartesian Coordinates for each atom. We transform these coordinates to an (rotation-invariant) Euclidean distance matrix and include the distance information as additional edge feature to the graph representation (More details in Appendix F).

**Graph Reconstruction and Generation**   We define a holdout set of 10,000 molecules and train the model on the rest. Up on convergence, we achieve on the hold out set a balanced accuracy of 99.93% for element type prediction, 99.99% for formal charge type prediction and 99.25% for edge type prediction (includs prediction of non-existence of edges). Distances between atoms are reconstructed with a root mean squared error of 0.33Å and a coefficient of determination of $R^2 = 0.94$.

| Dataset | PIGAE (ours) | ECFP |
|---|---|---|
| Classification (ROC-AUC ↑) | | |
| BACE | $0.824 \pm 0.005$ | $0.82 \pm 0.02$ |
| BBBP | $\mathbf{0.81 \pm 0.04}$ | $0.78 \pm 0.03$ |
| Regression (MSE ↓) | | |
| ESOL | $\mathbf{0.10 \pm 0.01}$ | $0.25 \pm 0.02$ |
| LIPO | $\mathbf{0.34 \pm 0.02}$ | $0.39 \pm 0.02$ |

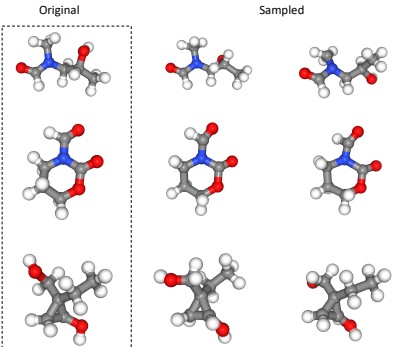

Table 3: Downstream performance for molecular property prediction tasks.

Figure 4: Example Molecular conformations sampled around original graph representation.

In the field of computational chemistry and drug discovery, the generation of energetically reasonable conformations for molecules is a topic of great interest [56]. Since our model is trained on approximating the data generating process for molecules and their conformations, we can utilize the trained model to sample molecular conformations. To retrieve a set of conformations, we encode a query molecule into its latent representation and randomly sample around this point by adding a small noise and decode the resulting representations. We utilize *Multidimensional Scaling* (MDS) [57] to transform a decoded distance matrix back to Cartesian Coordinates. In Figure 4, we show examples of molecular conformations sampled from the trained model. Under visual inspection, we find that sampled conformations differ from encoded conformations, while still being energetically reasonable (e.g., rotation along rotatable bonds while avoiding clashes between atoms).

**Molecular Property Prediction**   Finally, we evaluate the learned representations for molecular property prediction. In order to accurately represent molecular graphs from different parts of the chemical space, we train our proposed model on a larger dataset retrieved from the public PubChem database [58]. We extracted organic molecules with up to 32 heavy atoms, resulting into a set of approximately 67 million compounds (more details in Appendix F). We evaluate the representations of the trained model based on the predictive performance of a SVM on two classification and two regression tasks from the MoleculeNet benchmark [59]. We compare representations derived from our pre-trained model with the state-of-the-art Extended Connectivity Fingerprint molecular descriptors (radius=3, 2048 dim) [60]. For each task and descriptor, the hyperparameter $C$ of the SVM was tuned in a nested cross validation. The results are presented in Table 3. Descriptors derived from our pre-trained model seem to represent molecular graphs in a meaningful way as they outperform the baseline on average in three out of four tasks.

## 5   Conclusion, Limitations and Future Work

In this work we proposed a permutation invariant autoencoder for graph-level representation learning. By predicting the relation (permutation matrix) between input and output graph order, our proposed model can directly be trained on node and edge feature reconstruction, while being invariant to a distinct node order. This poses, to the best of our knowledge, the first method for non-contrastive and non-adversarial learning of permutation invariant graph-level representations that can also be used for graph generation and might be an important step towards more powerful representation learning methods on graph structured data or sets in general. We demonstrate the effectiveness of our method in encoding graphs into meaningful representations and evaluate its competitive performance in various experiments. Although we propose a way of reducing the computational complexity by only attending on incoming messages in our directed message self-attention framework, in its current state, our proposed model is limited in the number of nodes a graph can consist of. However, recently, much work has been done on more efficient and sparse self-attention frameworks [61, 62, 63]. In future work, we aim at building up on this work to scale our proposed method to larger graphs. Moreover, we will investigate further into the generative performance of the proposed model, as this work was mainly concerned with its representation learning capability.

## Funding Disclosures

D.A.C. received financial support from European Commission grant numbers 963845 and 956832 under the Horizon2020 Framework Program for Research and Innovation. F.N. acknowledges funding from the European Commission (ERC CoG 772230 "ScaleCell") and MATH+ (AA1-6). F.N. is advisor for Relay Therapeutics and scientific advisory board member of 1qbit and Redesign Science.

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
