# Appendix:
# Permutation-Invariant Variational Autoencoder for Graph-Level Representation Learning

**Robin Winter**
Bayer AG
Freie Universität Berlin
robin.winter@bayer.com

**Frank Noé**
Freie Universität Berlin
frank.noe@fu-berlin.de

**Djork-Arné Clevert**
Bayer AG
djork-arne.clevert@bayer.com

## Appendix A

**Propositions 1.** *A square matrix* $\mathbf{P}$ *is a real permutation matrix if and only if* $C(\mathbf{P}) = 0$ *and the doubly stochastic constraint* $p_{ij} \geq 0 \ \forall (i,j)$, $\sum_i p_{ij} = 1 \ \forall j$, $\sum_j p_{ij} = 1 \ \forall i$ *holds.*

**Proof.** *A permutation matrix is a doubly stochastic matrix with only one* 1 *in every row and column.*
($\Rightarrow$) *If P is a permutation matrix,* $C(\mathbf{P}) = 0$ *and the doubly stochastic constraint are satisfied by definition.*
($\Leftarrow$) *If* $C(\mathbf{P}) = 0 \Rightarrow \sum_i H(\bar{p}_{ij}) = 0 \ \forall j \ \wedge \ \sum_j H(\bar{p}_{ij}) = 0 \ \forall i$. *Thus, if doubly stochastic constrain are satisfied,* $\bar{p}_{ij}$ *can only have one non-zero element in each row* $i$ *and column* $j$ *equal to* 1.

**Remark** *Since we apply the row-wise softmax in Eq. (7),* $\sum_j p_{ij} = 1 \ \forall i$ *and* $p_{ij} \geq 0 \ \forall (i,j)$ *is always fulfilled. If* $C(\mathbf{P}) = 0$, *all but one entry in a column* $p_{i,\cdot}$ *are 0 and the other entry is 1. Hence,* $\sum_i p_{ij} = 1 \ \forall j$ *is fulfilled.*

## Appendix B

**Synthetic random graph generation**   To generate train and test graph datasets we utilized the python package *NetworkX* [1]. We sampled from ten different graph families with different parameter ranges, namely:

- Binominal graphs with edge probability $p \in (0.2, 0.6)$.
- Ego graphs extracted from Binominal graphs ($p \in (0.2, 0.6)$) selecting all neighbours of one random node.
- Watts-Strogatz small-world graphs with $k \in (2, 6)$ nearest neighbours and edge probability $p \in (0.2, 0.6)$.
- Newman-Watts-Strogatz small-world graphs with $k \in (2, 6)$ nearest neighbours and edge probability $p \in (0.2, 0.6)$.
- Random Regular graphs with degree $d \in (3, 6)$.
- Barabási–Albert graphs with $m \in (1, 6)$ edges preferentially attached to high degree nodes.
- Dual-Barabási–Albert graphs with $m_1 \in (1, 6)$ and $m_2 \in (1, 6)$ edges preferentially attached to high degree nodes and probability $p \in (0.1, 0.9)$ for sampling $m_1$ edges.

35th Conference on Neural Information Processing Systems (NeurIPS 2021).

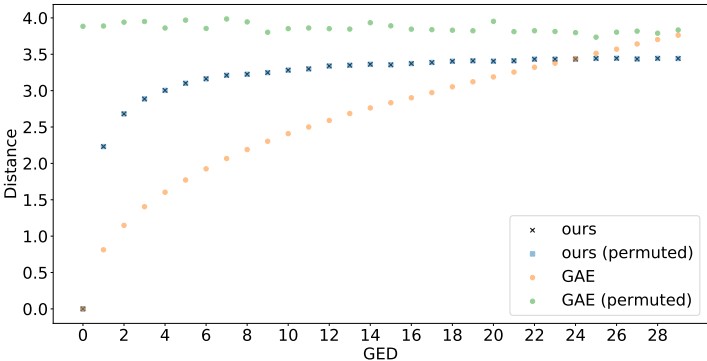

Figure 1: Distance in embedding space over increasing (approximated) graph editing distance (GED) average over 1000 Barabási–Albert graphs with 16 nodes. We compare our method against a classical graph autoencoder (GAE) for either the same graph ordering or random permutations of the same/edited graph.

- Extended-Barabási–Albert graphs with $m \in (1, 6)$ edges preferentially attached to high degree nodes and probability $p \in (0.1, 0.5)$ for adding an edge between existing nodes and probability $q \in (0.1, 0.5)$ for rewiring of existing edges.
- Random-Powerlaw-Tree graph with $\gamma = 3$.
- Random Geometric graph with edges between nodes in a unit square less than $r \in (0.35, 0.65)$ units apart.

## Appendix C

**Training Details**   We did not perform an extensive hyperparameter evaluation for the different experiments and mostly followed [2] for hyperparameter selection. We applied $L = 16$ layers of self-attention in both encoder and decoder, with a hidden dim chosen from $(256, 512)$ with either 16 or 32 attention heads with 64 hidden dimensions. Each self attention layer was followed by a point-wise fully connected neural network with two layers (1024 hidden dim) and a residual connection. We set the graph embedding dimension to 64. We tried different weightings of reconstruction and permutation matrix penalty loss to maximize the reconstruction accuracy with a discretized permutation matrix, while enabling stable training. In some settings we found it beneficial, to slowly decay the temperature constant $\tau$ of the Soft-Sort operator during training. We performed all experiments on a NVIDIA DGX-2 system, parallelizing models on up to 10 GPUs and training up to 5 days, depending on task and model size.

## Appendix D

**Graph Editing Distance**   In section 4.1 we describe how distances in the graph embedding space of our proposed model correlates with the graph editing distance (GED). One important property of the GED is its invariance to the node ordering of graphs that are compared. Thus, two similar graphs should be assigned the same small distance irregardless of the specific node ordering both graphs are represented in. Since embeddings produced by our proposed model are invariant to node permutations, distances between graphs in this embedding space, like the GED, are invariant to the node ordering as well. This is not the case for graph-level representations extracted by a classical graph autoencoder (GAE) [3]. In Figure 1 we plot the Euclidean distance between a reference graph representation and representations of graphs with increasing amount of variations applied to it. We followed the protocol described in Section 4.1 and created variations of graphs by successively substituting existing edges in the reference graph by new edges not present in the reference graph. We compare representations encoded by our method with representations encoded by a classical GAE.

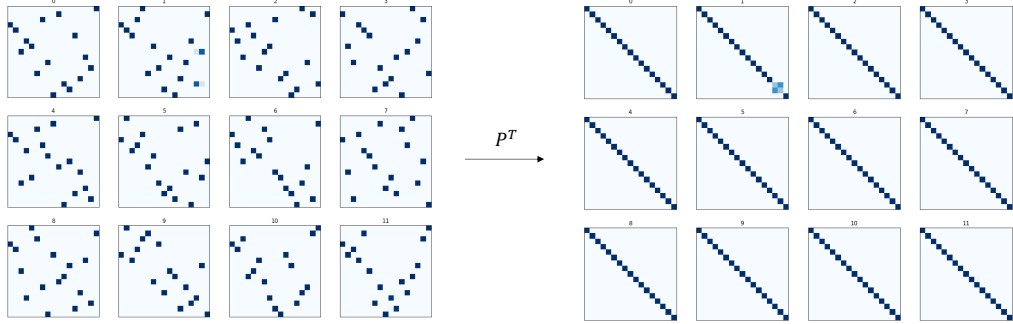

Figure 2: Left: Predicted permutation matrices for 12 random Barabási–Albert graphs. Right: Predicted permutation matrices after permuting the same input graphs by the transpose of the previously predicted permutation matrices.

We can see a clear correlation between the distance in embedding space and the amount of editing steps performed to the reference graph. However, in the case of the GAE, this correlation breaks, if we change the order of nodes in the edited graphs. This is expected, since the graph-level embedding of a GAE should permute equivalently with changes in the graph order. Since our graph-level representation is permutation-invariant, random permutations of the edited graphs have no effect on the Euclidean distance.

Interestingly, the distance of embeddings extracted by our method seem to converge after a certain amount of graph editing steps, while distances of embeddings extracted by the GAE seem to increase further. The reason for this might be that successively substituting edges of the reference graph with new edges must not necessarily result into more dissimilar graphs at some point. It is for instance possible to recreate the same graph again, just in a different node order. With increasing amounts of editing steps a graph should be on average as far away as the average distance between any pair of graphs (of the same size and family). In our case (Barabási–Albert graphs with m=4 and 16 nodes) this average pairwise distance between 1000 randomly generated graphs is approximately 3.3, which matches well with the value distances are converging against in Figure 1.

Also note, how random permutations of the same graph (isomorphic graphs) always resulted in a Euclidean distance of 0 and all graphs one editing step away (non-isomorphic) had a distance greater than 0.

## Appendix E

**Permutation Matrix**   As discussed in section 2.2 (Key architectural properties), we carefully designed our proposed model to make it invariant to permutations of the input graphs. By utilizing the permutation equivariant property of self-attention layers, the encoder and permuter model produce permutation matrices that equivalently permute with permutations in the input graph. In Figure 2, we show this for some example graphs (synthetic Barabási–Albert graphs). To better visually analyze the impact of the permutation of the input graph on the permutation matrix, we did the following: We first predicted permutation matrices for a set of random graphs (left hand side of Figure 2, note that we did not discretize these permutation matrices for this experiment) and then permuted the input graphs with permutation matrices that are defined by the transpose of these predicted permutation matrices. As a consequence, the predicted permutation matrices for these permuted graphs, are approximately (not discretized) equal to the identity matrix. This is exactly what we would expect. By permuting the input graphs by the transposed predicted permutation matrices, we brought the input graph in the canonical order learned by the decoder. Thus, no permutation is necessary and the permuter model predicts the identity matrix. Moreover, we can see how the permutation matrices equivalently permuted with the permutation of the input graph. A permutation of the input by $\mathbf{P}^T \mathcal{G}$ resulted in an equivalent permutation of the predicted permutation matrices $\mathbf{P}^T \mathbf{P} = \mathbb{1}$. Interestingly, in our experiments, this is not always the case. In Figure 2, image 1 shows one deviation from this property (not perfectly on the main diagonal). Inspecting the corresponding predicted permutation matrix (image 1 on the left hand side), we can see two entries (nodes) that are not assigned unambiguously,

resulting in bad approximation of a permutation matrix (which should only have one 1 in every row and column and zeros everywhere else). During inference, we would usually discretize such a permutation matrix to receive a valid permutation matrix. In fact, we find that for Barabási–Albert graphs with 20 nodes, 98 % of nodes have a value (confidence) assigned greater than 0.9. Thus, around 2 % of nodes cannot be assigned to the canonical order with a high confidence. Further investigation of these nodes has shown that they tend to have similar neighbourhoods. Approximately half of them are perfectly symmetric, meaning that they have the exact same neighbourhood, as Weisfeiler-Lehman graph hashing of their ego graphs revealed. Since very similar nodes (with respect to their neighbourhood) in a graph will receive similar node embeddings in the encoder model, the permuter model will score them similarly, resulting in ambiguous assignment in the predicted permutation matrix.

Still, graphs without such symmetric nodes, will receive a unambiguous permutation matrix. For such graphs we found in further experiments where we constructed for 10000 graphs 64 random permutations, that, as expected, the permutation matrix always permuted equivalently with the random permutation of the input graph.

## Appendix F

**Molecular graph datasets**   Molecular graphs were constructed utilizing following information for nodes:

- Atom type: 5 (C, N, O, F and H) for the QM9 dataset, 11 (C, N, O, F, S, Si, P, Cl, Br, I and H) for the PubChem dataset.
- Charge type: 3 (-1, 0, 1) for the QM9 dataset, 5 (-2, -1, 0, 1, 2) for the PubChem dataset.
- Ring membership

and following information for the edges:

- Bond type: 4 (single-, double-, triple- and aromatic bond) + 1 (no bond)
- Topological distance: normalized topological distance between two connected nodes.
- Euclidean Distance: For the QM9 dataset we transformed Cartesian coordinates of atoms to a normalized Euclidean distance matrix and used each entry as an additional edge feature.