# OpenReview forum: "Permutation-Invariant Variational Autoencoder for Graph-Level Representation Learning"
_NeurIPS.cc/2021/Conference — NeurIPS 2021 Poster_

### Official Review · Reviewer_d88Y · 2021-07-06

**Rating:** 6
**Confidence:** 2

**Summary:**

The authors proposed a permutation-invariant VAE for graph-structured data. The proposed model indirectly learns to match the node ordering of the input/output graph. The authors demonstrated the effectiveness of the model on graph reconstruction/generation tasks.

Overall, I found the paper both interesting and well-written. The contributions of this work are to show that The proposed work proposed permutation-invariant graph-level representation learning which can be used for graph generating modeling.

**Ethics Review Area:**

["I don’t know"]

**Main Review:**

I have a few questions/concerns about the work that I would like clarity on:

What is the computational limitation in the number of nodes in the graph? It would be interesting to add computational complexity with varying the number of nodes and comparison between the proposed model and other approaches.

Herein the authors adopt the idea of MPNN and represent graphs by its messages between nodes. How does the proposed approach handle self-loop (i.e., an edge connects a vertex to itself).

One of the key properties of the proposed work is that the enoder self-attention layers are invariant to permutations of the input node order. As the decoder is also based on permutation equivalant self-attention layers, this canonical order is completely defined and thus correctly calculates the reconstruction loss. This might be related to improve training stability compared to the other approach. It would be helpful if the authors can address this.

In general, VAEs can suffer from an issue known as latent variable collapse (or KL loss vanishing), where the posterior
collapses to the prior, and the model will ignore the latent codes in generative tasks. In the graph-level representation learning task, does the proposed approach have any issue related to latent variable collapse?


**Time Spent Reviewing:**

3

---

> ### Author Response · Authors · 2021-08-09
> **Author Response**
>
> We thank the reviewer for their comments and the encouraging assessment of our work. Below you will find our responses to your comments.
>
> ### Comment 1:
>
> „What is the computational limitation in the number of nodes in the graph? It would be interesting to add computational complexity with varying the number of nodes and comparison between the proposed model and other approaches.“
>
> #### Response:
>
> In section 2.3 (line 137-150) we mention the complexity with respect to the number of nodes, which is O(n^3). Currently, this is the main limitation of our proposed model as discussed in section 5. However, the aim of this work was to propose a model that, for the first time, can autoencode graphs to and from a single-vector permutation invariant graph-level representation and left optimization and scaling to larger graphs (in this work we only consider graphs up to 32 nodes) for future work. We believe that with the recent surge of interest in the Transformer community to develop more efficient attention mechanisms we can utilize those results, making our proposed model more efficient.
>
> ### Comment 2:
>
> “Herein the authors adopt the idea of MPNN and represent graphs by its messages between nodes. How does the proposed approach handle self-loop (i.e., an edge connects a vertex to itself).“
>
> #### Response:
>
> In our proposed message matrix representation self-loops are diagonal entries of the message matrix $m_{i,i}$ and are effectively representing the nodes itself. That way, we can represent the whole graph by one matrix, including both node and edge features. In this work we did not consider self-loops in the original graph, however, they could easily be integrated as additional node-features encoded in Equation (9).
>
>
> ### Comment 3:
>
> “One of the key properties of the proposed work is that the enoder self-attention layers are invariant to permutations of the input node order. As the decoder is also based on permutation equivalant self-attention layers, this canonical order is completely defined and thus correctly calculates the reconstruction loss. This might be related to improve training stability compared to the other approach. It would be helpful if the authors can address this.“
>
> #### Response:
>
> It is true that as a key aspect of our proposed model we exploit the invariant and equivariant property of the chosen encoder and decoder architecture. But the canonical order of the decoded graph is not defined because of that. It is learned by the decoder (without supervision) and aligned to the sequence of position embeddings. To correctly calculate the reconstruction loss we have to permute this canonical order to the present order of the input. This permutation operation is learned by the permuter model. As the decoder is permutation equivariant we can simply permute the position embeddings. We are not sure how this can be related to the training stability and compared to other approaches. We are sorry, but we think that we might not understand this comment correctly. Could the reviewer clarify if we did not answer sufficiently?
>
> ### Comment 4:
> “In general, VAEs can suffer from an issue known as latent variable collapse (or KL loss vanishing), where the posterior collapses to the prior, and the model will ignore the latent codes in generative tasks. In the graph-level representation learning task, does the proposed approach have any issue related to latent variable collapse?”
>
> #### Response:
>
> Indeed, one has to careful in choose the right (weighted) combination of the different loss contributions (reconstruction, KLD, permutation matrix entropy). When we trained our proposed graph VAE and e.g. chose the kld loss contribution to outweigh the permutation matrix entropy loss (Equation (8)), the posterior can collapses to the prior and information is passed through the permutation matrix leading to bad performance during inference (where no permutation matrix is provided). If weighted correctly, the latent variable does not collapse, as information has to be passed through the latent code to minimize the reconstruction loss.

---

### Official Review · Reviewer_394F · 2021-07-13

**Rating:** 4
**Confidence:** 5

**Summary:**

This paper proposes a new graph variational auto-encoder that is invariant to permutation of node orders. The authors design a permuter model that can predict a permutation of node orders in order to better measure the distance of reconstructed graph and input graph in a permutation-invariant manner. To verify the approach, extensive experiments are conducted on both synthetic and real-world molecular graph datasets.

**Limitations And Societal Impact:**

See my main review for limitations of this work.

**Main Review:**

The paper is well motivated. Endowing the representation/generative model for graph-structured data with permutation-invariant property is interesting and significant. However, the proposed method lacks enough novelty and theoretical soundness.

First, the technical contribution of  this work is not clear. There are quite a few existing works having considered permutation-invariant property for graph/set representation learning and proposing effective solutions, e.g. [1-3]. Therefore, the authors' claim "This is the first to consider permutation invariance in graph-level representation" is not true.

Furthermore, given this fact, the seemingly novel part of this work lies in the permuter model that is trained to predict a permutation of node orders. However, this method is quite straightforward but lacks enough theoretical justification. With the permuter output P, the orignal ELBO objective would be changed and it is not clear what distribution of P is considered in this work and how its variance of sampling would impact the output. Also, the learning objective cannot guarantee that the learned P is a desired one. Here are two concerns: 1) since the learning for P and learning for decoder are coupled and there are no ground-truth information for P, the learning of two modules would possibly interfered with each other and lead to undesirable solution; 2) the learned P would possibly degenerate to trivial solutions, e.g. P becomes an identity matrix which makes no difference for learning permutation invariance.

Finally, the comparison with existing models is not sufficient. There are quite a few existing works on graph edit distance and permutation-invariant representation learning and they are not discussed and compared by the authors. Also, the experiment on molecular property prediction is not convincing with no comparison with other GNN-based models.


[1] An efficient algorithm for graph edit distance computation, Chen et. al, 2019
[2] Generative Adversarial Set Transformers, Stelzner et. al, 2020
[3] Deep Set Prediction Networks,  Zhang et. al, 2020

**Time Spent Reviewing:**

5

---

> ### Author Response · Authors · 2021-08-09
> **Author Response**
>
> We thank the reviewer for their comments. Below you will find our responses to your comments.
>
> ### Comment 1:
>
> “First, the technical contribution of this work is not clear. There are quite a few existing works having considered permutation-invariant property for graph/set representation learning and proposing effective solutions, e.g. [1-3]. Therefore, the authors' claim "This is the first to consider permutation invariance in graph-level representation" is not true.“
>
> #### Response:
>
> Our claim is more restricted as put by the reviewer. We claim: “To the best of our knowledge, our work proposes the first approach for non-contrastive permutation invariant graph-level representation learning, that can also be used for graph generative modeling“.
>
> The works cited by the reviewer propose either
> -	permutation invariant graph-level representation encoder [1] or
> -	permutation invariant graph decoder [2]
>
> but not the combination of both or work on sets [3]. Hence, we argue that our claim is true. Still, we would like to add the reference to the related work discussion and thank the reviewer for these suggestions.
>
>
> ### Comment 2:
>
> “Furthermore, given this fact, the seemingly novel part of this work lies in the permuter model that is trained to predict a permutation of node orders. However, this method is quite straightforward but lacks enough theoretical justification. With the permuter output P, the orignal ELBO objective would be changed and it is not clear what distribution of P is considered in this work and how its variance of sampling would impact the output. Also, the learning objective cannot guarantee that the learned P is a desired one. Here are two concerns: 1) since the learning for P and learning for decoder are coupled and there are no ground-truth information for P, the learning of two modules would possibly interfered with each other and lead to undesirable solution; 2) the learned P would possibly degenerate to trivial solutions, e.g. P becomes an identity matrix which makes no difference for learning permutation invariance.
> „
>
>
> #### Response:
> In section 2.1 we motivate and justify the use of the permuter model as an extension to the original ELBO objective in order to define a proper like likelihood for graphs that are permutation invariant. As the likelihood for non-trivial graphs can only be maximization if the right permutation matrix is predicted, the loss can only be minimized if desired permutation matrices P are learned. Regarding your concerns: 1) Of course, in practice it is always harder to jointly train multiple models compared to a single model with a single objective. But in this work, we motivate why it is necessary in order to train a permutation invariant graph autoencoder. The same argument could be made against other significant work too (e.g. GANs) and is misleading. 2) If P degenerate to a trivial solution the reconstruction loss could not be minimized, as the input and output graph (node features and adjacency matrix) would not match. From this concern it seems the reviewer might have misunderstood a crucial part of our proposed method: The decoder has only access to the permutation invariant graph-level embedding. It has no access to the node ordering of the input graph. Thus, a proper permutation matrix is needed to align input and output graph order to minimize the reconstruction loss.
>
> ### Comment 3:
>
> „Finally, the comparison with existing models is not sufficient. There are quite a few existing works on graph edit distance and permutation-invariant representation learning and they are not discussed and compared by the authors. Also, the experiment on molecular property prediction is not convincing with no comparison with other GNN-based models.“
>
> #### Respond:
>
> In the related work section, we discuss recent state-of-the art contrastive learning approaches and compare our method against InfoGraph in table 2. We agree that we should discuss works on graph edit distance as an alternative contrastive learning approach (i.e., SimGNN, Bai et al.) and would like to add this to our current manuscript. Thank you for this suggestion. We did not compare against supervised GNN models in the molecular property prediction experiment, as we are mainly concerned about unsupervised-learned representations in this work.

---

### Official Review · Reviewer_V8HM · 2021-07-15

**Rating:** 5
**Confidence:** 2

**Summary:**

This paper focuses on unsupervised learning of graph-level representations --- achieved here through a VAE. One of the main hurdles in this problem is to ensure the permutation invariance of the graph: the order in which the nodes appear in the network should not influence the graph embedding, which typically creates issues in minimizing a reconstruction loss between encoded and decoded graph. This paper tackles the problem by proposing a graph autoencoder architecture that is invariant to the ordering of nodes in a graph and that learns a permutation  along side the graph embedding --- thereby simplifying the problem of optimizing the reconstruction loss. The encoder/decoder parts of the algorithms use a message passing scheme to learn messages across each edge (concatenation of node features and edge features) with attention mechanisms. This is however quite an expensive procedure --- complexity of O(n^3).

Relationship to other works: Other works have tried to use approximate graph matching technique using the permutation matrix that maximizes the similarity, with is O(n4) --- expensive. Previous approaches have also already tried to deploy generative models for graphs, but they only achieve permutation equivariance (and not permutation invariance).

Potential Impact: This is an interesting problem, since, as the authors highlight, the main focus of the recent literature has been on supervised learning at the node or graph-level. Unsupervised methods for graphs and generative models have the potential to unlock many possibilities in generating new graph samples, new chemical compounds, etc.

Contribution:
The pipeline is a concatenation of tricks developed in other settings: VAE + permutation matrix + message passing with attention. The way the authors learn the permutation matrix is new though.

**Limitations And Societal Impact:**

None --- except global warming because of potentially intensive computational costs.

**Main Review:**

Pros:
- This is an interesting and difficult problem, which hasn't been as much in the limelight of the community, and the authors do contribute to the discussion by proposing a new method --- although the main contribution here seems to be the learning of the permutation matrix.

Cons:
- The method is extremely computationally expensive. That could be fine --- there is not so much done in the literature, so it is important to  start somewhere and the authors were indeed able to generate new molecules on 32 nodes or so.
- I am more concerned in the potential of the experiments to show for the performance and relative superiority of the model over existing methods:
      - It is important to sum up what the complexity of the method should be (total cost). It would have been interesting to also show the running time for all the experiments, (a) for comparison with other methods, and (b) because it is an expensive method and that any user trying their luck with it could really benefit from insights into how much time they should expect the computations to run.
      - The complexity of the method is extremely high: the authors are fitting a learned encoder/decoder with attention (so many many parameters), a permutation method, and they are additionally learning how many nodes to generate. This creates concerns regarding the convergence of the method (this seems to be very highly non convex, so how sure are we that the convergence will be well behaved?)
     - It would have been interesting to compare this set of methods against more classical alternatives; The classification task seems quite good to evaluate the performance of the method, so it might have been interesting (on top of methods looking at distances between original input graphs) to see what the other GAE+Graphite methods yield, as well as using more traditional statistical methods (ie, ERGMS) for completeness of comparison.
     - Since the method is so complex, I am not sure that the negative log-likelihood is a fair comparison between all the different methods. Negative LL is typically used in a bayesian setting to compare models with the same complexity --- which is not the case here. What should perhaps be computed instead is either an DIC/AIC/BIC, or  better yet, since this is a non linear model, taking inspiration from the Bayesian literature, and computing  Posterior Predictive ordinates or Conditional Predictive Ordinates...


To sum up, this could be a potential interesting avenue of research, but considering the complexity of the method, I am not sure that the current experiments draw a convincing enough picture to show that the costs are worth the effort.


**Time Spent Reviewing:**

2h30

---

> ### Author Response · Authors · 2021-08-09
> **Author Response**
>
> We thank the reviewer for their comments. Below you will find our responses to your comments and proposed changes to our manuscript according to your valuable suggestions.
>
> ### Comment 1:
>
> „The method is extremely computationally expensive. That could be fine --- there is not so much done in the literature, so it is important to start somewhere and the authors were indeed able to generate new molecules on 32 nodes or so.“
>
> ### Reponse:
>
> This is correct and we pointed out this limitation in the conclusion/limitations section. We believe, however, that with the recent surge of interest in the Transformer community to develop more efficient attention mechanisms we can utilize those results, making our proposed model more efficient and scaling it up to larger graphs. We share the reviewer’s assessment, that it is important to start somewhere, demonstrating the properties of our proposed framework and leave the optimization of the model for future work.
>
> ### Comment 2:
>
> “I am more concerned in the potential of the experiments to show for the performance and relative superiority of the model over existing methods:
> - It is important to sum up what the complexity of the method should be (total cost). It would have been interesting to also show the running time for all the experiments, (a) for comparison with other methods, and (b) because it is an expensive method and that any user trying their luck with it could really benefit from insights into how much time they should expect the computations to run.
> - The complexity of the method is extremely high: the authors are fitting a learned encoder/decoder with attention (so many many parameters), a permutation method, and they are additionally learning how many nodes to generate. This creates concerns regarding the convergence of the method (this seems to be very highly non convex, so how sure are we that the convergence will be well behaved?)”
>
> #### Response:
>
> We mention the training time and resources used in the supplementary. It is true, that the proposed model has extremely high complexity - as most transformer-like architectures have today. We saw training times of a few hours one a single GPU for small synthetic graph datasets up to weeks on multi-GPU machines for large molecular graphs. However, in our experiments we did not experience bad learning behavior. Moreover, as demonstrated in the field of NLP, it is often only required to pretrain such a huge model once on a large dataset (e.g. of molecules) and reuse its weights to finetune it to other specific tasks. Still, to make the comparison to other method possible, we will add a more detailed list of resource costs for each experiment to supplementary. Thank you for this suggestion.
>
> ### Comment 3:
>
> “It would have been interesting to compare this set of methods against more classical alternatives; The classification task seems quite good to evaluate the performance of the method, so it might have been interesting (on top of methods looking at distances between original input graphs) to see what the other GAE+Graphite methods yield, as well as using more traditional statistical methods (ie, ERGMS) for completeness of comparison.“
>
> #### Response:
>
> This is a very good point. We motivate our work by claiming that that pooled local node-level representations (e.g. as learned by a classical GAE/Graphite) are inferior to a global graph-level representation (as learned by our proposed model or contrastive learning approaches). Hence, we should include these methods in the graph classification tasks (Table 2). We repeated the experiment including embeddings extracted by a classical GAE model (averaged node representations) trained on the same dataset as our model (and InfoGraph) and trained a Support Vector Machine on predicting the different graph classes, following the same hyperparameter optimization scheme. This model achieved an average accuracy of 0.65 +/- 0.01, significant worse compared to our proposed model (0.83 ± 0.01). We think this experiment is an import addition to our manuscript, supporting our claims that pooled local node-level representations are inferior to a global graph-level representation for such tasks as discussed above. We would like to include these results and an additional discussion (of pooled local node-level vs global graph-level representations) to the manuscript and thank the reviewers for making us aware of this missing justification/discussion.
>
> ### Comment 4:
>
> „Since the method is so complex, I am not sure that the negative log-likelihood is a fair comparison between all the different methods. Negative LL is typically used in a bayesian setting to compare models with the same complexity --- which is not the case here. What should perhaps be computed instead is either an DIC/AIC/BIC, or better yet, since this is a non linear model, taking inspiration from the Bayesian literature, and computing Posterior Predictive ordinates or Conditional Predictive Ordinates...“
>
> #### Response:
>
> It is true, that our proposed model has much higher complexity (multi-layer self-attention) and we mainly choose NNL to compare to the results reported in the Graphite paper. If we consider number of parameters in the metric (DIC/AIC/BIC) the classical GAE would most likely come up at the top. However, increasing the number of parameters in the GAE did also not increase its performance. In this work we were mostly concerned with investigating if a single-vector permutation-invariant graph-level representation can be learned and not how efficient (in the number of parameters) we can make it. As mentioned in the conclusion, we will leave this for future work. Still, we acknowledge the reviewer’s concern and would like to mention the difference in model complexity in the discussion. We thank the reviewer for making us aware of this.

---

### Official Review · Reviewer_1wGW · 2021-07-16

**Rating:** 6
**Confidence:** 4

**Summary:**

This work proposes PIGAE, a graph-level VAE with a permutation-invariant reconstruction loss (expected likelihood). To achieve permutation-invariance, the authors propose learning a (relaxed) permutation as a function of the encoded (input) graph. The paper demonstrates the performance of PIGAE on synthetic benchmarks, and real-world molecular graphs.


**Limitations And Societal Impact:**

The authors properly discuss the limitations and the societal impacts.

**Main Review:**

In general, the reconstruction loss for graph-level VAEs is not invariant to permutations. As a consequence, the decoders can overfit to specific node orderings observed during training. Previous works end up using (sometimes costly) heuristics to tackle this issue. To overcome this limitation, the authors learn the permutation as a function of the encoded graph. Then, the authors use this learned permutation to make their loss permutation invariant. The results on graph reconstruction are particularly impressive. Overall, the authors touch on an important problem, and the work is novel. Notably, the authors fail to clearly flesh out the challenge in training graph-level VAEs (neither in the abstract nor introduction). As a result, the text may be difficult to follow for a general audience.

**Comments and questions**

* For any graph, the mean $\mu$ and the variance $\sigma$ of the latent $z$ are linear functions of $m_{0,0}$ (a scalar). This implies that the mean of $z$'s for all encoded graphs lies in the same line. The same applies to $\sigma$'s. This seems to be a strong restriction.

* To evaluate the generation of PIGAE, the authors perturb the latent of an existing graph and decode it back (Figure 3). This explores only a small part of the support of the prior. Consequently, I believe the experiment does properly evaluate the quality of PIGAE samples. To assess the generation quality of PIGAE, the authors should simply decode samples from the prior. Overall, the paper lacks examples of graphs generated by PIGAE for different problems/datasets.

* To sample from VAEs, we draw a sample from the prior and propagate it through the decoder. In this work, the decoder requires a permutation. How does PIGAE work during sampling? Is the permutation set to some default value?

* Proposition 1 seems equivalent to stating 'permutation matrices have columns and lines with entropy equal to zero'. This seems rather trivial, and I'm not sure if a proposition is needed here.

* After proposition 1, the authors discuss graph isomorphism and state that PIGAE "can solve the graph isomorphism problem at least for all graphs it can reconstruct". Since there is no characterization of 'graphs it can reconstruct', this sentence looks vague.

* Regarding literature coverage, the authors miss some works on normalizing flows for graphs (e.g., [1] and [2])

[1]: Graph Normalizing Flows, NeurIPS 2019

[2]: MoFlow: An Invertible Flow Model for Generating Molecular Graphs, KDD 2020

**Time Spent Reviewing:**

6

---

> ### Author Response · Authors · 2021-08-09
> **Author Response**
>
> We thank the reviewer for their comments. Below you will find our responses to your comments and proposed changes to our manuscript according to your valuable suggestions.
>
> ### Comment 1:
>
> “For any graph, the mean μ and the variance σ of the latent z are linear functions of m0,0 (a scalar). This implies that the mean of z's for all encoded graphs lies in the same line. The same applies to σ's. This seems to be a strong restriction.
>
> #### Response:
>
> We would like to clarify, that $m_{0,0}$ is not a scalar but a vector. The message matrix M is tensor with dimensions [n, n, d], with the number of nodes n and number of message features d (see line 127-136). We hope this addresses the reviewer’s concern sufficiently.
>
> ### Comment2:
>
> “To evaluate the generation of PIGAE, the authors perturb the latent of an existing graph and decode it back (Figure 3). This explores only a small part of the support of the prior. Consequently, I believe the experiment does properly evaluate the quality of PIGAE samples. To assess the generation quality of PIGAE, the authors should simply decode samples from the prior. Overall, the paper lacks examples of graphs generated by PIGAE for different problems/datasets.“
>
> #### Response:
>
> The reviewer correctly points out that we do not evaluate the generative properties of our method a lot in this work. We are aware of this and, as pointed out in the conclusion, leave a thorough evaluation of the generative performance for future work. However, we acknowledge that the generative performance could be interesting for the general audience, and we would like to add graphs (e.g. synthetic graphs) sampled from the prior to the supplementary. Since submission of the paper, we also investigated the ability to interpolate between graphs, by linearly interpolating their embeddings in the latent space. We think this could be interesting for the graph generation community, since e.g. in a classical autoencoder framework (without a permutation invariant embedding), there is no clear way how to interpolate between graphs, as graphs would need to be aligned first.
> Since we are not able to upload figures to open review, we uploaded a depiction of an example interpolation between two graphs with a description of the experiment on figshare (private anonymous link: https://figshare.com/s/ee86702825d67aa959c1). To the best of our knowledge, such a graph interpolation in the latent space has not been report in other work yet and might show similar impact in the graph generation community as interpolation of latent spaces did in the NLP or image filed.
>
> ### Comment 3:
>
> “To sample from VAEs, we draw a sample from the prior and propagate it through the decoder. In this work, the decoder requires a permutation. How does PIGAE work during sampling? Is the permutation set to some default value?“
>
> ### Reponse:
>
> This is an interesting question, as it touches on a crucial part of our proposed model that (apparently) is probably not clearly enough explained in our current manuscript. We would like to emphasize that the decoder does not require a permutation to function. The decoder learns its own canonical ordering that is aligned to the position embeddings added to the initialized message matrix. We emphasize, that the permuter model (i.e. the permutation of the position embeddings) is not necessary for the decoding process in itself. We only need it to align input and output graphs to calculate the loss. During inference/sampling, i.e., using the trained decoder as generative model, we no longer need/use the permuter model (we do not permute the position embeddings).
> We hope this answers the reviewer’s question and would like to add an additional explanation of this property to the method section.
>
> ### Comment 4:
>
> “Proposition 1 seems equivalent to stating 'permutation matrices have columns and lines with entropy equal to zero'. This seems rather trivial, and I'm not sure if a proposition is needed here.“
>
> #### Response:
>
> This is true and we were also not sure if we had to include this to properly justify the additional loss term to force the permutation matrices to be valid. If the reviewers agree, we are also happy to remove the Proposition or move it to the supplementary.
>
> ### Comment 5:
>
> “After proposition 1, the authors discuss graph isomorphism and state that PIGAE "can solve the graph isomorphism problem at least for all graphs it can reconstruct". Since there is no characterization of 'graphs it can reconstruct', this sentence looks vague.
>
> #### Response:
>
> As discussed in this paragraph, we argue that we can detect isomorphic graphs, as they will have the same embedding, since the encoder is permutation invariant. However, we must consider the possibility, that two non-isomorphic graphs could wrongly be encoded to the same embedding. Since, the decoder is a deterministic function, the same embedding will produce the same decoded graph up to a permutation as defined by the predicted permutation matrix. Thus, if both graphs have the same embedding z and can be perfectly reconstructed by the decoder, they must be isomorphic, as they can only differ in their node ordering. Consequently, if they can be perfectly reconstructed by the decoder but have a different embedding, they must be non-isomorphic.
> Therefore, we use the term “for all graphs it can reconstruct“, because we cannot be sure for the rest. “reconstruct” means in this context that input and output graph can be perfectly matched, i.e., zero reconstruction loss. We are happy to add an additional sentence is this line of argument to explain this more clearly. Thank you for pointing this out.
>
> ### Comment 6:
>
> “Regarding literature coverage, the authors miss some works on normalizing flows for graphs (e.g., [1] and [2])
> [1]: Graph Normalizing Flows, NeurIPS 2019
> [2]: MoFlow: An Invertible Flow Model for Generating Molecular Graphs, KDD 2020“
>
> #### Repsonse:
>
> This is correct and we thank the reviewer for these suggestions. We will add them into the related work discussion.

---

### Decision · Program_Chairs · 2021-09-28

**Decision:**

Accept (Poster)

**Comment:**

Two reviewers recommend rejecting the submission and two recommending accepting the submission. Reviewer 394F, who recommends rejecting the submission, appears to have misunderstood key aspects of the article (i.e., why the permutation matrix P will not become the identity). Therefore, I am not basing my recommendation of that reviewer’s rating. My main concern about this submission is the comprehensiveness of the experimental results. It’s not clear from them whether/when the proposed method will outperform similar methods like GAE and Graphite. In the authors’ response some additional comparisons to GAE have been provided, which seem encouraging, but are not comprehensive.

**Consistency Experiment:**

NeurIPS has a long history of experimentation. In 2014, NeurIPS ran an experiment in which 10% of submissions were reviewed by two independent committees to quantify the randomness in the review process. This year, we repeated a variant of this experiment to see how the quality of the review process has changed over time.  This paper was part of the experiment and was therefore assigned to two committees (consisting of reviewers, an Area Chair, and a Senior Area Chair) that reached independent decisions.  If both committees made the same recommendation, this recommendation was followed. If a single committee recommended acceptance, the paper was accepted (with the exception of a few cases in which the other committee identified what we considered a fatal flaw, e.g., an error in a key result).

This copy’s committee reached the following decision: **Reject**

The other committee assigned to the paper recommended **Accept (Poster)**.  You can find the other set of reviews, along with any follow up discussion with the authors here:
https://openreview.net/forum?id=pTmYjQadg9